# BlendGAN: Implicitly GAN Blending for Arbitrary Stylized Face Generation

**Mingcong Liu**
Y-tech, Kuaishou Technology
liumingcong03@kuaishou.com

**Qiang Li**[*]
Y-tech, Kuaishou Technology
liqiang03@kuaishou.com

**Zekui Qin**
Y-tech, Kuaishou Technology
qinzekui03@kuaishou.com

**Guoxin Zhang**
Y-tech, Kuaishou Technology
zhangguoxin@kuaishou.com

**Pengfei Wan**
Y-tech, Kuaishou Technology
wanpengfei@kuaishou.com

**Wen Zheng**
Y-tech, Kuaishou Technology
zhengwen@kuaishou.com

## Abstract

Generative Adversarial Networks (GANs) have made a dramatic leap in high-fidelity image synthesis and stylized face generation. Recently, a layer-swapping mechanism has been developed to improve the stylization performance. However, this method is incapable of fitting arbitrary styles in a single model and requires hundreds of style-consistent training images for each style. To address the above issues, we propose BlendGAN for arbitrary stylized face generation by leveraging a flexible blending strategy and a generic artistic dataset. Specifically, we first train a self-supervised style encoder on the generic artistic dataset to extract the representations of arbitrary styles. In addition, a weighted blending module (WBM) is proposed to blend face and style representations implicitly and control the arbitrary stylization effect. By doing so, BlendGAN can gracefully fit arbitrary styles in a unified model while avoiding case-by-case preparation of style-consistent training images. To this end, we also present a novel large-scale artistic face dataset AAHQ. Extensive experiments demonstrate that BlendGAN outperforms state-of-the-art methods in terms of visual quality and style diversity for both latent-guided and reference-guided stylized face synthesis. Our project webpage is https://onion-liu.github.io/BlendGAN/

## 1   Introduction

Generative adversarial networks (GANs) [1] have achieved impressive performance in various tasks such as image generation [2, 3, 4, 5, 6, 7, 8, 9], image super-resolution [10, 11], and image translation [12, 13, 14, 15, 16, 17, 18, 19]. In recent years, GAN has also been widely used for face stylization such as portrait drawing [20], caricature [21, 22], manga [23], and anime [24]. With the deepening of GAN research, the community has further paid attention to the quality of generated images and the disentanglement ability of latent space.

To achieve high-quality generation and latent disentanglement, StyleGAN [8] introduces the adaptative instance normalization layer (AdaIN) [25] and proposes a new generator architecture, which

---

[*]Corresponding author.

35th Conference on Neural Information Processing Systems (NeurIPS 2021).

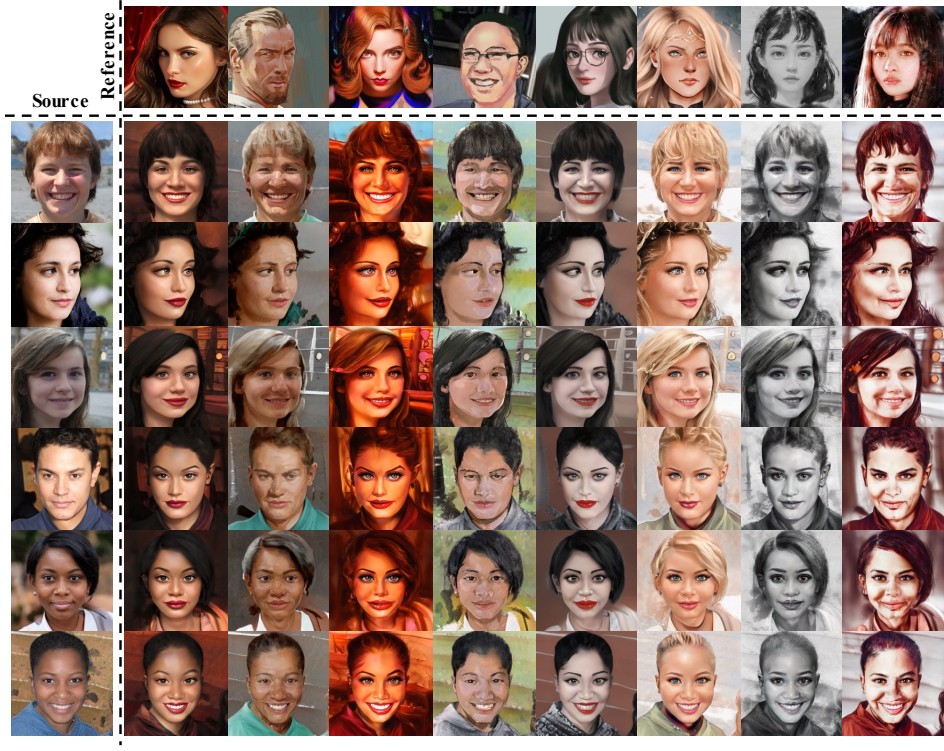

Figure 1: Illustration of some reference-guided synthesis results. Our framework can generate stylized face images with high quality.

achieves pretty good performance in generating face images. StyleGAN2 [9] explores the causes of droplet-like artifacts and further improves the quality of generated images by redesigning generator normalization. StyleGAN2-ada [26] proposes an adaptive discriminator augmentation mechanism which reduces the required number of images to several thousands. In the experiment of StyleGAN2-ada, the model is trained on MetFaces dataset [26] to generate artistic faces, however, the style of image is uncontrollable, and the corresponding original face cannot be obtained.

Recently, a layer-swapping mechanism [27] has been proposed to generate high-quality natural and stylized face image pairs, and equipped with an additional encoder for unsupervised image translation [28]. In particular, these methods first finetune StyleGAN from a pretrained model with target-style faces and then swap some convolutional layers with the original model. Given randomly sampled latent codes, the original model generates natural face images, while the layer-swapped model outputs stylized face images correspondingly. Though effective at large, these methods have to train models in a case-by-case flavor, thus a single model can only generate images with a specific style. Besides, these methods still require hundreds of target-style images for finetuning, and one has to carefully select the number of training iterations, as well as the swapped layers.

In this paper, we present BlendGAN for arbitrary stylized face generation.[2] BlendGAN resorts to a flexible blending strategy and a generic artistic dataset to fit arbitrary styles without relying on style-consistent training images for each style. In particular, we analyze a stylized face image as composed of two latent parts: a face code (controlling the face attributes) and a style code (controlling the artistic appearance). Firstly, a self-supervised style encoder is trained via an instance discrimination objective to extract the style representation from artistic face images. Secondly, a *weighted blending module* (WBM) is proposed to blend the face and style latent codes into a final code which is then fed into a generator. By controlling the indicator in WBM, we are able to decide which parts of the face and style latent codes to be blended thus controlling the stylization effect. By combining the

---

[2]Our framework can also cooperate with GAN inversion [29, 30, 31, 32, 33] or StyleGAN distillation [34] methods to enable end-to-end style transfer or image translation. We will show the results in the appendix.

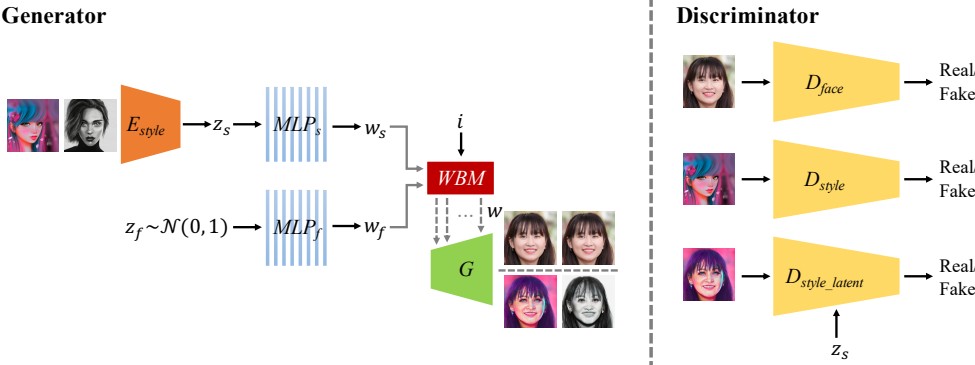

Figure 2: Overview of the proposed framework. The style encoder $E_{style}$ extracts the style latent code $\mathbf{z}_s$ of a reference style image. The face latent code $\mathbf{z}_f$ is randomly sampled from the standard Gaussian distribution. Two MLPs transform face and style latent codes into their $W$ spaces separately, then they are combined by the *weighted blending module (WBM)* and fed into generator $G$ to synthesise natural and stylized face images. Three discriminators are used in our method. The face discriminator $D_{face}$ distinguishes between real and fake natural-face images, the style discriminator $D_{style}$ distinguishes between real and fake stylized-face images, and the style latent discriminator $D_{style\_latent}$ predicts whether the stylized-face image is consistent with the style latent code $\mathbf{z}_s$.

style encoder and the generator with WBM, our framework can generate natural and stylized face image pairs with either a randomly sampled style or a reference style (see Figure 1).

As for the generic artistic data, we present a novel large-scale dataset of high-quality artistic face images, Artstation-Artistic-face-HQ (AAHQ), which covers a wide variety of painting styles, color tones, and face attributes. Experiments show that compared to state-of-the-art methods, our framework can generate stylized face images with higher visual quality and style diversity for both latent-guided and reference-guided synthesis.

## 2 Related Work

**Generative Adversarial Networks** The recent years have witnessed rapid advances of generative adversarial networks (GANs) [1] for image generation. The key to the success of GANs is the adversarial training between the generator and discriminator. Various improvements to GANs have been proposed to stabilize GAN training and improve the image quality. WGAN [3] uses the Wasserstein distance as a new cost function to train the network. WGAN-GP [4] and SNGAN [5] improve the stability of GAN by introducing gradient penalty and spectral normalization separately to make the training satisfy the 1-Lipschitz constraint. SAGAN [35] enlarges the receptive field using the self-attention mechanism. BigGAN [6] shows the significant improvement of image quality when training GANs at large scales. StyleGAN [8, 9] redesigns the generator architecture with AdaIN layers, making it better disentangle the latent factors of variation. In this work, we employ GANs to generate high-quality stylized face images.

**Image-to-Image Translation** With the good performance of GANs, many GAN-based image-to-image translation techniques have been explored in recent years [12, 13, 14, 16, 17, 36, 18, 37, 19, 38]. For image translation in two domains, Pix2pix [12] proposes the first unified image-to-image framework to translate images to another domain. CycleGAN [13] proposes a cycle-consistency loss, making the network train with unpair data. UNIT [14] maps images in source and target domains to a shared latent space to perform unsupervised image translation. UGATIT [24] proposes an adaptive layer-instance normalization (AdaLIN) layer to control shapes during translation. For image translation in multi-domains, MUNIT [16] extends UNIT to multi-modal contexts by decomposing images into content and style spaces. FUNIT [17] leverages a few-shot strategy to translate images using a few reference images from a target domain. DRIT++ [18] also proposes a multi-modal and multi-domain model using a discrete domain encoding. Training with both latent codes and reference images, StarGANv2 [19] could provide both latent-guided and reference-guided synthesis. Park et.al.

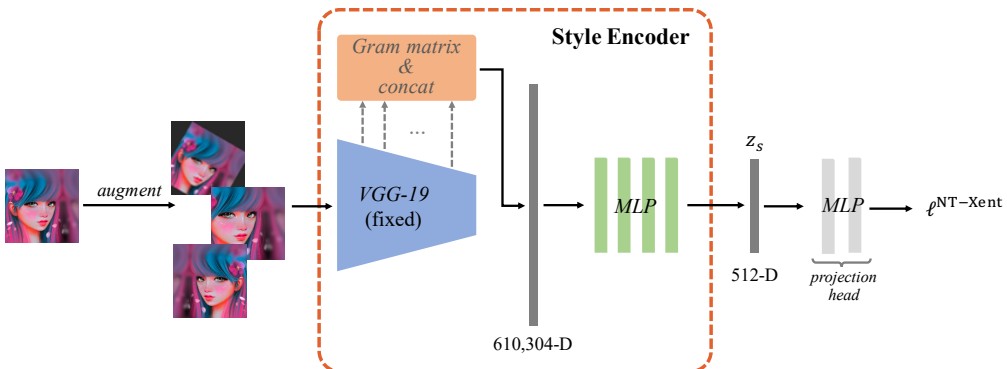

Figure 3: Architecture of the proposed style encoder, which consists of a pretrained *VGG-19* [46] network, a *Gram matrix and concat* module and an MLP predictor. More detailed notations are described in the contexts of Sec. 3.1.

[38] utilize the autoencoder structure to swap the style and content of two given images, and it can also be regarded as a reference-guided synthesis method. Although these methods could translate images to multiple target domains, the diversity of target domains is still limited and these methods require hundreds of training images for each domain. Besides, the above methods could only operate at a resolution of at most 256x256 when applied to human faces.

**Neural Style Transfer**    The arbitrary stylized face generation task could also be regarded as a kind of the neural style transfer [39, 40, 41, 25]. Gatys *et al.* [39], for the first time, proposes a neural style transfer method based on optimization strategy. Johnson *et al.* [40] trains a feed-forward network to avoid the time consumption of optimization. Luan *et al.* [42] proposes a photorealism regularization term to preserve the structure of source images. The above methods could only transfer source images to a single style with a model, many works extend the style flexibility and achieved multi-style or arbitrary style transfer such as MSG-Net [41], StyleRemix [43], AdaIN [25], WCT [44] and SANet [45]. Although arbitrary style transfer methods could transfer images to arbitrary styles, they only transfer the reference style to source images globally without considering local semantic styles, which leads to global texture artifacts in the outputs.

## 3 Method

Our goal is to train a generator

$$\hat{x}_f, \hat{x}_s = G(\mathbf{z}_f, \mathbf{z}_s, i), \tag{1}$$

that can generate a image pair (natural face image $\hat{x}_f$ and its artistic counterpart $\hat{x}_s$) with a pair of given latent code $(\mathbf{z}_f, \mathbf{z}_s)$ and a blending indicator $i$. The face latent code $\mathbf{z}_f$ controls the face identity, and the style latent code $\mathbf{z}_s$ controls the style of $\hat{x}_s$. The blending indicator $i$ decides which parts of $\mathbf{z}_f$ and $\mathbf{z}_s$ are blended to generate plausible images. Figure 2 illustrates an overview of our framework.

### 3.1 Self-supervised Style Encoder

The style encoder $E_{style}$ is independently trained to extract the style representation from an artistic face image while discarding the content information such as face identity. Many style transfer methods [39, 40, 41] obtain the style representation by calculating the Gram matrices of feature maps extracted by a pretrained VGG [46] network. However, if simply concatenating the Gram matrices of all the selected feature maps, the resulting style latent code ('style embedding') would have a very high dimension. For example, if we choose layers $relu1\_2$, $relu2\_2$, $relu3\_4$, $relu4\_4$ and $relu5\_4$, the style embedding would be a 610,304-dimensional vector $(64^2 + 128^2 + 256^2 + 512^2 * 2)$. Higher latent code dimension leads to sparser distribution, which makes the generator harder to train and perform worse in style disentanglement. Hence, following the self-supervised method SimCLR [47, 48], we train the style encoder to reduce the dimension of style embeddings.

Figure 3 illustrates the architecture of our style encoder. The style image is augmented and then sent to a pretrained VGG [46] network. We calculate the Gram matrices of selected feature maps, flatten and concatenate them into a long vector (610,304-D in our case). Then the vector is passed into a 4-layer MLP module (predictor) to reduce the dimension and generate the style embedding $\mathbf{z}_s$ of the input style image. Following $\mathbf{z}_s$, another 2-layer MLP performs as a projection head for contrastive learning. The augmentations in the original SimCLR include affine transformation and color transformation, following the assumption that affine and color transformations do not change the classes of objects. However, for style encoding, the image style is strongly related to color, so we only use affine transformations in the augmentation step. During training the style encoder, only parameters of two MLP modules are optimized while *VGG-19* is fixed to perform as a feature extractor. The network is trained using NT-Xent loss [47]:

$$\ell_{i,j}^{\mathrm{NT-Xent}} = -\log \frac{\exp(\mathrm{sim}(\mathbf{z}_i, \mathbf{z}_j)/\tau)}{\sum_{k=1}^{2N} \mathbb{1}_{[k \neq i]} \exp(\mathrm{sim}(\mathbf{z}_i, \mathbf{z}_k)/\tau)}, \tag{2}$$

where $(i, j)$ is a pair of positive examples (augmented from the same image), $\mathbf{z}$ is the projection result, $\mathrm{sim}(\cdot, \cdot)$ is cosine similarity between two vectors, and $\tau$ is a temperature scalar.

## 3.2 Generator

The generator $G$ is similar to StyleGAN2 [9], but its input is a combination of two latent codes - face latent code $\mathbf{z}_f$ and style latent code $\mathbf{z}_s$. In particular, $\mathbf{z}_f$ controls the face identity of the generated images, and is randomly sampled from the standard Gaussian distribution $\mathcal{N}(0, 1)$. $\mathbf{z}_s$ controls the style of the generated stylized image, it could be either randomly sampled from $\mathcal{N}(0, 1)$ or encoded by the style encoder $E_{style}$ from a reference artistic image. Two MLPs transform $\mathbf{z}_f$ and $\mathbf{z}_s$ into their $W$ spaces as $\mathbf{w}_f$ and $\mathbf{w}_s$ separately. For $1024 \times 1024$ output, the dimension of $\mathbf{w}_f$ and $\mathbf{w}_s$ are all $18 \times 512$. Controlled by the blending indicator $i$, the *weighted blending module (WBM)* combines $\mathbf{w}_s$ and $\mathbf{w}_f$ into $\mathbf{w}$, which is then fed into the generator as the final code.

**WBM**  Different resolution layers in the StyleGAN model are responsible for different features in the generated image [8] (e.g. low resolution layers control face shape and hair style, high resolution layers control color and lighting). Therefore, the blending weights of $\mathbf{w}_s$ and $\mathbf{w}_f$ should not be consistent at different layers. We propose *WBM* to blend the two codes, which can be described as:

$$\mathbf{w} = \mathbf{w}_s \odot \hat{\alpha} + \mathbf{w}_f \odot (\mathbb{1} - \hat{\alpha}), \tag{3}$$

$$\hat{\alpha} = \alpha \odot \mathbf{m}(i), \tag{4}$$

$$\mathbf{m}(i; \theta) = [m_0, m_1, m_2, \ldots, m_n], \ m_j = \begin{cases} 0 & j < i \\ \theta & j = i \ , \quad \theta \in (0, 1), \\ 1 & j > i \end{cases} \tag{5}$$

where $\odot$ denotes the broadcasting element-wise product, $\mathbf{w}_s, \mathbf{w}_f \in \mathbb{R}^{18 \times 512}$ are the input codes, $\alpha \in \mathbb{R}^{18}$ is a learnable vector to balance the blending weights for $\mathbf{w}_s$ and $\mathbf{w}_f$ in different layers. $\mathbf{m}(i; \theta)$ has the same dimension as $\alpha$, it controls which layer should be blended. If $i = 0$, $\mathbf{w} = \mathbf{w}_s \odot \alpha + \mathbf{w}_f \odot (\mathbb{1} - \alpha)$, the generator outputs stylized face image $\hat{x}_s$; if $i = 18$, $\mathbf{w} = \mathbf{w}_f$, the generator outputs natural face image $\hat{x}_f$. When $0 < i < 18$, some low resolution layers are not influenced by style codes, which ensures the $\hat{x}_s$ to keep the same face identity as $\hat{x}_f$ (see Figure 2). $\theta$ is used for finer adjustment.

## 3.3 Discriminator

There are three discriminators in our framework(see the right column in Figure 2). The face discriminator $D_{face}$ and the style discriminator $D_{style}$ have the same architecture as in StyleGAN2 [9]. $D_{face}$ distinguishes between real and fake natural-face images, it only receives images sampled from the natural-face dataset (FFHQ) or generated by $G$ when $i = 18$. In the contrast, $D_{style}$ distinguishes between real and fake stylized-face images and it only receives images sampled from the artistic face dataset (AAHQ) or generated by $G$ when $i = 0$. The style latent discriminator $D_{style\_latent}$ has the same architecture as the *projection discriminator* [49], which has two inputs - a generated stylized-face image $\hat{x}_s$ and a style latent code $\mathbf{z}_s$, it predicts whether $\hat{x}_s$ is consistent with $\mathbf{z}_s$. During training the network, we use an embedding queue to storage the style latent codes of the previously sampled style images embedded by $E_{style}$, and randomly sample one as the fake $\mathbf{z}_s$.

## 3.4 Training objectives

Given a natural-face image $x_f \in \mathcal{X}_{natural}$, a stylized-face image $x_s \in \mathcal{X}_{style}$ and a randomly sampled face latent code $\mathbf{z}_f \sim \mathcal{N}(0,1)$, we train our framework using the following adversarial objectives.

For the face discriminator, the blending indicator $i = 18$. $G$ only takes $\mathbf{z}_f$ as the input and learns to generate a natural-face image $\hat{x}_f = G_{i=18}(\mathbf{z}_f)$ via an adversarial loss

$$\mathcal{L}_{face} = \mathbb{E}_{x_f}[\log D_{face}(x_f)] + \mathbb{E}_{\mathbf{z}_f}[\log(1 - D_{face}(G_{i=18}(\mathbf{z}_f)], \tag{6}$$

where $D_{face}(\cdot)$ denotes the output of the face discriminator.

For the style discriminator, the blending indicator $i = 0$. $G$ takes $\mathbf{z}_f$ and $x_s$ as inputs, and generate a stylized-face image $\hat{x}_s$ which has the same style as $x_s$. The loss is described as

$$\mathcal{L}_{style} = \mathbb{E}_{x_s}[\log D_{style}(x_s)] + \mathbb{E}_{\mathbf{z}_f, x_s}[\log(1 - D_{style}(G_{i=0}(\mathbf{z}_f, E_{style}(x_s))))], \tag{7}$$

where the style encoder $E_{style}$ extracts the style latent code of $x_s$, and $D_{style}(\cdot)$ denotes the output of the style discriminator.

For the style latent discriminator, we denote $\mathbf{z}_s = E_{style}(x_s)$ as the style latent code of $x_s$, and randomly sample another style latent code $\mathbf{z}_s^-$ from the embedding queue as a negative sample, then the loss could be described as

$$\mathcal{L}_{style\_latent} = \mathbb{E}_{\mathbf{z}_f, x_s}[\log D_{style\_latent}(x_s, \mathbf{z}_s)] + \mathbb{E}_{\mathbf{z}_f, x_s}[\log(1 - D_{style\_latent}(\hat{x}_s, \mathbf{z}_s^-))]. \tag{8}$$

Consequently, we combine all the above loss functions as our full objective as follows:

$$\mathcal{L}_G = \mathcal{L}_{face} + \mathcal{L}_{style} + \mathcal{L}_{style\_latent}, \tag{9}$$

$$\mathcal{L}_D = -\mathcal{L}_G. \tag{10}$$

# 4 Experiments

Our proposed method is able to generate arbitrary stylized face images with high quality and diversity. In this section, we describe evaluation setups and test our method both qualitatively and quantitatively on a large amount of images spanning a large range of face and style varieties.

**Datasets**  As described in Sec. 3, for arbitrary stylized face generation, we need a natural-face dataset and an artistic face dataset to train the networks. We use FFHQ [8] as the natural-face dataset, which includes 70,000 high-quality face images[3]. In addition, we build a new dataset of artistic-face images, Artstation-Artistic-face-HQ (AAHQ), consisting of 33,245 high-quality artistic faces at $1024^2$ resolution (Figure 4). The dataset covers a wide variety in terms of painting styles, color tones, and face attributes. The artistic images are collected from Artstation[4] (thus inheriting all the biases of that website) and automatically aligned and cropped as FFHQ. Finally, we manually remove images without faces or with low quality. More details of this dataset can be found in the appendix.

**Baselines**  We compare our model with several leading baselines on diverse image synthesis including AdaIN [25], MUNIT [16], FUNIT [17], DRIT++ [18], and StarGANv2 [19]. All of these methods could synthesise images in different modalities through the control of latent codes or reference images. All the baselines are trained on FFHQ and AAHQ using the open-source implementations provided by the authors, and our BlendGAN code is based on an unofficial PyTorch implementation of StyleGAN2[5].

**Evaluation metrics**  To evaluate the quality of our results, we use Frechet inception distance (FID) metric [50] to measure the discrepancy between the generated images and AAHQ dataset. A lower FID score indicates that the distribution of generated images are more similar to that of AAHQ dataset and the generated images are more plausible as real stylized-face images. In addition, we

---

[3]The FFHQ dataset is under Creative Commons BY-NC-SA 4.0 license.

[4]https://www.artstation.com

[5]https://github.com/rosinality/stylegan2-pytorch

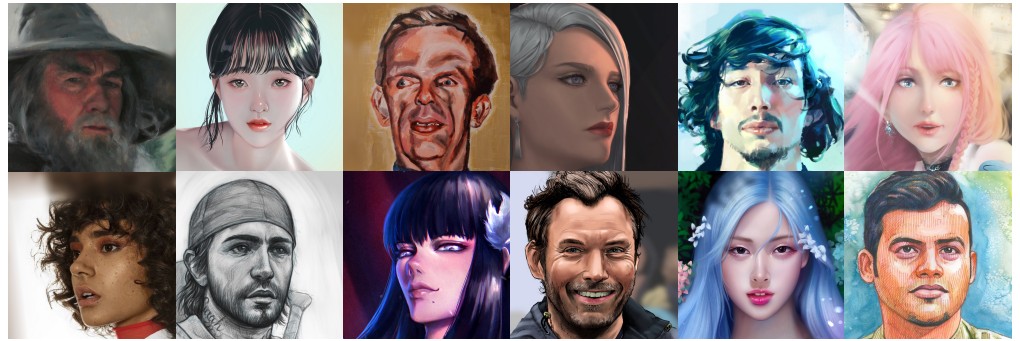

Figure 4: The AAHQ dataset offers a lot of variety in terms of painting styles, color tones and face attributes.

Table 1: FID and LPIPS comparison with different blending indicators.

| Indicator $i$ | | 0 | 2 | 4 | 6 | 8 | 10 | 12 | 14 | 16 | 18 |
|---|---|---|---|---|---|---|---|---|---|---|---|
| latent | FID $\downarrow$ | **8.97** | 12.45 | 12.75 | 23.17 | 42.45 | 57.34 | 68.31 | 76.91 | 78.25 | 76.73 |
| | LPIPS $\uparrow$ | **0.581** | 0.571 | 0.568 | 0.515 | 0.459 | 0.367 | 0.304 | 0.207 | 0.145 | 0.159 |
| reference | FID $\downarrow$ | **3.79** | 6.39 | 6.82 | 15.08 | 34.33 | 51.49 | 63.73 | 76.00 | 77.11 | 76.97 |
| | LPIPS $\uparrow$ | **0.661** | 0.651 | 0.650 | 0.599 | 0.540 | 0.450 | 0.377 | 0.237 | 0.191 | 0.160 |

adopt the learned perceptual image patch similarity (LPIPS) metric [51] to measure the style diversity of generated images. Following [19], as for a face latent code (corresponding to a natural-face image), we randomly sample 10 style latent codes or reference style images to generate 10 outputs and evaluate the LPIPS scores between every 2 outputs. We randomly sample 1000 face latent codes, repeat the above process and average all of the scores as the final LPIPS score. A higher LPIPS score indicates that the model could generate images with larger style diversity.

### 4.1 Blending indicator

As described in Sec. 3.2, the indicator $i$ in WBM controls in which layers the weights of $\mathbf{w}_s$ and $\mathbf{w}_f$ should be blended. In StyleGAN generator, low-resolution layers are responsible for face shape and high-resolution layers control color and lighting. Hence, as $i$ gets larger, less low-resolution layers are influenced by style codes, and the face shapes of stylized outputs are more similar to the natural-face images. As shown in Figure 5, when $i = 0$, the generated images have the strongest style as well as the most different face identities from the corresponding natural-face images. When $i = 18$, the generated images completely lose the style information and become natural-face images. Table 1 also shows that the style quality and diversity of the generated images decrease with the increase of $i$. We notice that when $i = 6$, the generated images have a good balance between stylization strength and face shapes consistency with the natural-face images. To this end, we set $i = 6$ in the qualitative experiments, while we use both $i = 0$ and $i = 6$ in the quantitative experiments.

### 4.2 Comparison on arbitrary stylized face generation

Our framework can generate stylized-face images with two mechanisms: latent-guided generation and reference-guided generation.

**Latent-guided generation.** For latent-guided generation, the style latent code $\mathbf{z}_s$ is randomly sampled from the standard Gaussian distribution. Figure 6 shows a qualitative comparison of the competing methods. The results of AdaIN [25] are not shown because it is not able to take latent codes as the style inputs. Since FUNIT is designed for few-shot image-to-image translation, it could not generate face images when inputting randomly sampled style latent codes. The results of MUNIT and DRIT++ have severe visible artifacts and they cannot reserve face identities in some images. StarGANv2 has the best performance among the baselines but the generated images still have some

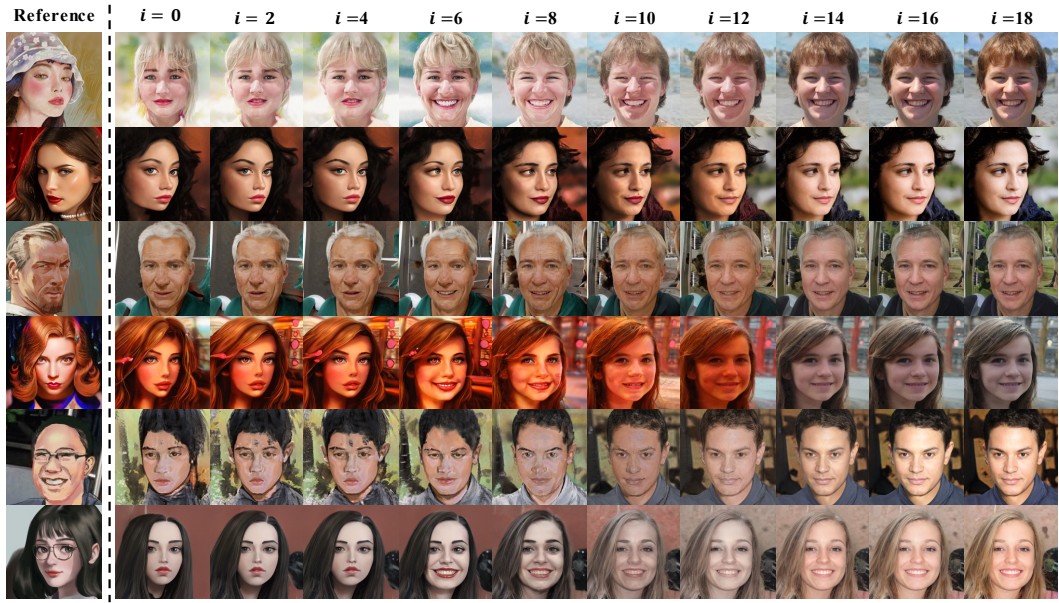

| Reference | $i = 0$ | $i = 2$ | $i = 4$ | $i = 6$ | $i = 8$ | $i = 10$ | $i = 12$ | $i = 14$ | $i = 16$ | $i = 18$ |

Figure 5: Reference-guided stylized-face images with different blending indicators.

Table 2: Quantitative comparison on latent-guided stylized-face generation.

| Method | FID $\downarrow$ | LPIPS $\uparrow$ |
|---|---|---|
| MUNIT | 29.69 | 0.394 |
| FUNIT | 176.96 | 0.500 |
| DRIT++ | 22.53 | 0.448 |
| StarGANv2 | 50.20 | 0.312 |
| **BlendGAN** ($i = 6$) | 23.17 | 0.515 |
| **BlendGAN** ($i = 0$) | **8.97** | **0.581** |

Table 3: Quantitative comparison on reference-guided stylized-face generation.

| Method | FID $\downarrow$ | LPIPS $\uparrow$ |
|---|---|---|
| AdaIN | 37.23 | 0.345 |
| MUNIT | 103.61 | 0.192 |
| FUNIT | 87.71 | 0.327 |
| DRIT++ | 31.71 | 0.241 |
| StarGANv2 | 50.03 | 0.307 |
| **BlendGAN** ($i = 6$) | 15.08 | 0.599 |
| **BlendGAN** ($i = 0$) | **3.79** | **0.661** |

subtle artifacts and are not as natural as our results. We observe that the generated stylized-face images of our method have the highest visual quality, and well preserve the face identities of the source images. Quantitative comparison is shown in Table 2. Although when $i = 6$, the FID score of our method is slightly higher than DRIT++, the qualitative comparison in Figure 6 shows that our stylized images have less artifacts and higher visual quality. When $i = 0$, our method has the lowest FID score and the highest LPIPS, indicating that our model could generate stylized images with the best visual quality and style diversity.

**Reference-guided generation.** For reference-guided generation, the style latent code $\mathbf{z}_s$ is embedded by the style encoder $E_{style}$ from a reference artistic face image. Figure 7 illustrates qualitative comparison results. The results show that the style transfer method AdaIN can not consider semantic style information and introduces severe visible artifacts in their generated stylized faces. FUNIT and DRIT++ also lead to visible artifacts in some images, as shown in the first and fourth rows. Although MUNIT allows to input a reference image at the testing stage, it only transfers the average color of the reference images, and the results do not have similar textures to the references. StarGANv2 has the best performance among the baselines, however, the style of some generate images is not consistent with their references, as shown in the fourth and fifth rows. As a comparison, our BlendGAN generates stylized-face images with the highest quality and the style of generated images is the most consistent with the references thanks to our well-designed training strategy. Quantitative comparison is shown in Table 3. For $i = 0$ and $i = 6$, our method achieves FID of 3.79 and 15.08, which outperform all the previous leading methods by a large margin. This implies that images generated by our model are the most similar to real stylized-face images. The LPIPS of our method is also

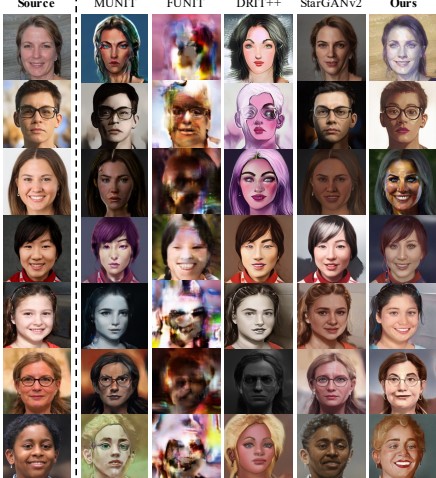
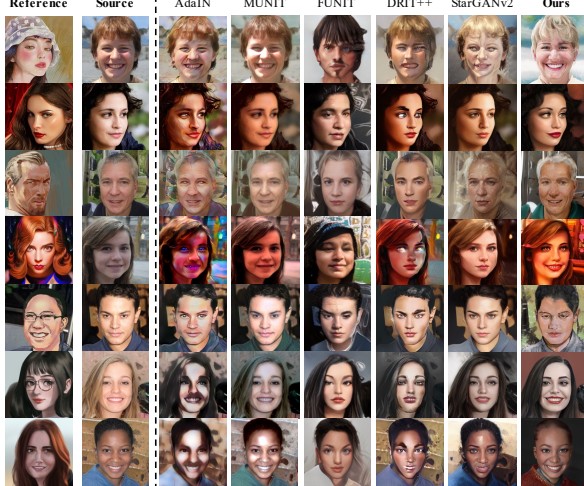

Figure 6: Comparison of latent-guided generation. The source images are generated by our model with indicator $i = 18$, and the style latents are randomly sampled from $\mathcal{N}(0, 1)$. From left to right: the source image, the results of MUNIT [16], FUNIT [17], DRIT++ [18], StarGANv2 [19] and our BlendGAN. Digital zoom-in recommended.

Figure 7: Comparison of reference-guided generation. The source images are generated by our model with indicator $i = 18$, and the reference images are sampled from the AAHQ dataset. From left to right: the reference artistic-face image, the source natural-face image, the results of AdaIN [25], MUNIT [16], FUNIT [17], DRIT++ [18], StarGANv2 [19] and our BlendGAN. Digital zoom-in recommended.

the highest, indicating that our model can produce faces with the most diverse style considering the reference images. Besides, it is worth noting that our method with $i = 0$ performs better than $i = 6$ because low-resolution layers are not affected by indicator $i$.

## 5    Conclusions

In this paper, we propose a novel framework for arbitrary stylized face generation. Training with FFHQ and a new large-scale artistic face dataset (AAHQ), our framework can generate infinite arbitrary stylized faces as well as their unstylized counterparts, and the generation could be either latent-guided or reference-guided. Specifically, we propose a self-supervised style encoder to extract the style representation from reference images, as well as a weighted blending module to implicitly combine the face and style latent codes for controllable generation. The experimental results show that our proposed model could generate images with high quality and style diversity, and the generated images have good style consistency with reference images, remarkably outperforming the previous leading methods. Our experiments are currently only performed on face datasets, future work includes extending this framework to other domains, such as stylized landscape generation.

## Broader Impact

This work would be of great interest to the general community. It shows the great potential of applying GAN models to stylized image generation, which could be further used for other tasks, such as style transfer, style manipulation, and new style creation.

Although the goal of this work is to generate faces for stylistic purposes, it still has some ethical challenges like other face generation methods. It is well documented that common face datasets lack diversity [52]. As our model is trained using the FFHQ [8] dataset, it inherits the biases of this dataset like other StyleGAN-based methods [27, 28, 53]. Specifically, similar to those discussed in the previous works [54, 53], for natural-face images, the model generates light-skinned faces more frequently than darker-skinned faces, and this concern can be mitigated by enlarging the racial diversity of the natural-face dataset. For stylized-face images, our AAHQ dataset already covers a

wide diversity of faces (see Figure 4 and Section B of the supplementary materials). In addition, since our framework only extracts the style representation of images in the AAHQ dataset and does not extract information related to face identity, the ethical biases (if any) of this dataset will not be transferred to the generated images. Therefore, the stylized-face images generated by our model have equally good performance for both light-skinned and darker-skinned faces (as shown in Figures 1, 6 and 7), which further indicates the superiority of our framework. Nonetheless, our model still has the same challenges as PULSE [53] (i.e., input images with darker skin are now stylized as lighter-skinned faces in the output), and these challenges are worthy of further research.

Besides, though not the purpose of this work, the natural-face images generated by our model could also be misused in the "deepfakes" relevant applications, such as fake portraits in social media [55]. Wang et al. [56] showed that a classifier trained to classify between real images and synthetic images generated by ProGAN [7], was able to detect fakes produced by other generators, among them, StyleGAN [8] and StyleGAN2 [9]. This finding can partially mitigate the above concern.

Last but not least, faces in the FFHQ and our AAHQ datasets are biometric data and thus need to be sourced and distributed more carefully, with attention to potential privacy, consent and copyright issues.

## Acknowledgments and Disclosure of Funding

We sincerely thank all the reviewers for their comments. We also thank Xin Miao for helpful discussions as well as the implementation [57] of the projection discriminator [49]. Besides, we thank Zhenyu Guo for help in preparing the comparison to StarGANv2 [19]. No external funding was received for this work.

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
