# BlendGAN: Implicitly GAN Blending for Arbitrary Stylized Face Generation
## Supplementary Materials

**Mingcong Liu**
Y-tech, Kuaishou Technology
liumingcong03@kuaishou.com

**Qiang Li**[*]
Y-tech, Kuaishou Technology
liqiang03@kuaishou.com

**Zekui Qin**
Y-tech, Kuaishou Technology
qinzekui03@kuaishou.com

**Guoxin Zhang**
Y-tech, Kuaishou Technology
zhangguoxin@kuaishou.com

**Pengfei Wan**
Y-tech, Kuaishou Technology
wanpengfei@kuaishou.com

**Wen Zheng**
Y-tech, Kuaishou Technology
zhengwen@kuaishou.com

## A    Training Details

The training code of our framework is modified from the unofficial PyTorch implementation of StyleGAN2[2], and we keep most of the hyper-parameters unchanged. We use the Adam [1] optimizer with $\beta_1 = 0$, $\beta_2 = 0.99$, and the learning rate is set to 0.002.

For the style encoder $E_{style}$, we use the AAHQ dataset downsampled to $256 \times 256$ resolution. The two MLP modules are composed of FC layers with equalized learning rates and leaky ReLU activations with $\alpha = 0.2$. In the augmentation step, we only use affine transformations and not color transformations to preserve the image style. The style encoder is trained independently for about one week on 128M images with four Tesla V100 GPUs using a minibatch size of 256 (500k iterations).

For the generator and the three discriminators, we use the FFHQ [2] and AAHQ datasets with $1024 \times 1024$ resolution. The weights of the generator $G$ and the face discriminator $D_{face}$ are loaded from the official pretrained StyleGAN2 model to reduce the training time, while those of the style discriminator $D_{style}$ and style latent discriminator $D_{style\_latent}$ are randomly initialized. The generator and the discriminators are jointly optimized with non-saturating logistic loss and path length regularization, and they are trained for four days on 1.6M images with four Tesla V100 GPUs using a minibatch size of 8 (200k iterations).

## B    The AAHQ dataset

We build a new dataset, AAHQ, by collecting artistic-face images from Artstation[3]. Dataset images are collected from the "portraits" channel of the website and automatically aligned as the FFHQ dataset. We manually remove images without faces or with low quality, and finally produce 33,245 high-quality artistic faces at $1024^2$ resolution. Figure 1 illustrates the t-SNE visualization of AAHQ, which shows a large style diversity of face images.

---

[*]Corresponding author.
[2]https://github.com/rosinality/stylegan2-pytorch
[3]https://www.artstation.com

35th Conference on Neural Information Processing Systems (NeurIPS 2021).

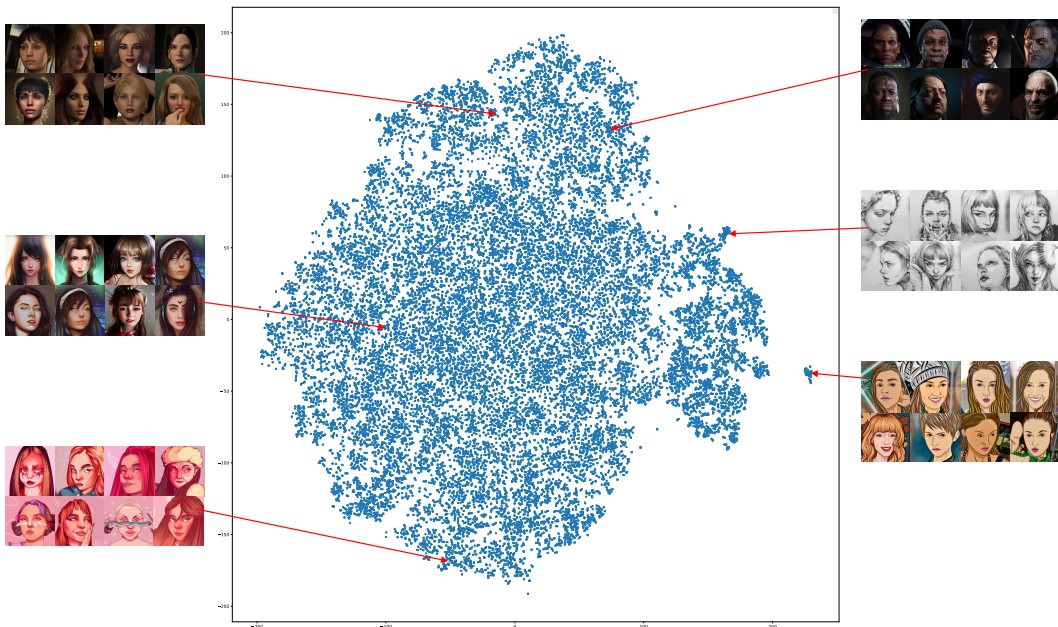

Figure 1: t-SNE visualization of the AAHQ dataset. The style embeddings of images are encoded by our proposed style encoder.

## C  Arbitrary Style Transfer

Many GAN inversion methods have been proposed to encode a given image back into the latent space of a pretrained GAN model [3, 4, 5, 6, 7], and our framework can synthesise corresponding arbitrary stylized images of a given face latent code. Hence, cooperating with GAN inversion methods, our framework is able to achieve arbitrary style transfer of a given face image. We use pSp [7], a learning-based GAN inversion method, to extract the latent code of face images. During training the pSp framework, we set the indicator $i$ of our BlendGAN generator to 18, making it only generate natural face images. We only optimize weights of the pSp encoder, while weights of the generator are fixed, and the total framework is trained to reconstruct the input images. Figure 2 and Figure 3 show the latent-guided and reference-guided style transfer results of given face images, respectively, which demonstrate that our BlendGAN can generate arbitrary stylized face images with face latent codes extracted by a pSp encoder.

## D  Ablation Study: Style Encoder

We conduct an experiment to compare our self-supervised style encoder with the method that directly concatenating the Gram matrices as the style representations of reference images. As described in Sec. 3.1 of the main text, the directly concatenating method leads to a 610,304-dimensional vector, which is used as the style latent code $\mathbf{z}_s$ and fed into the generator. Comparison results are shown in Figure 4. When $i = 0$, all the layers of the generator are influenced by the style latent code. Result images of the directly concatenating method have similar face identities and head poses to their reference images, which means that this method leaks content information of reference images to style latent codes. In contrast, images generated by our method when $i = 0$ only have the same style as references, and their face identities as well as head poses are similar to corresponding natural faces ($i = 18$), indicating that our style encoder can effectively extract the style representations of references while removing their content information.

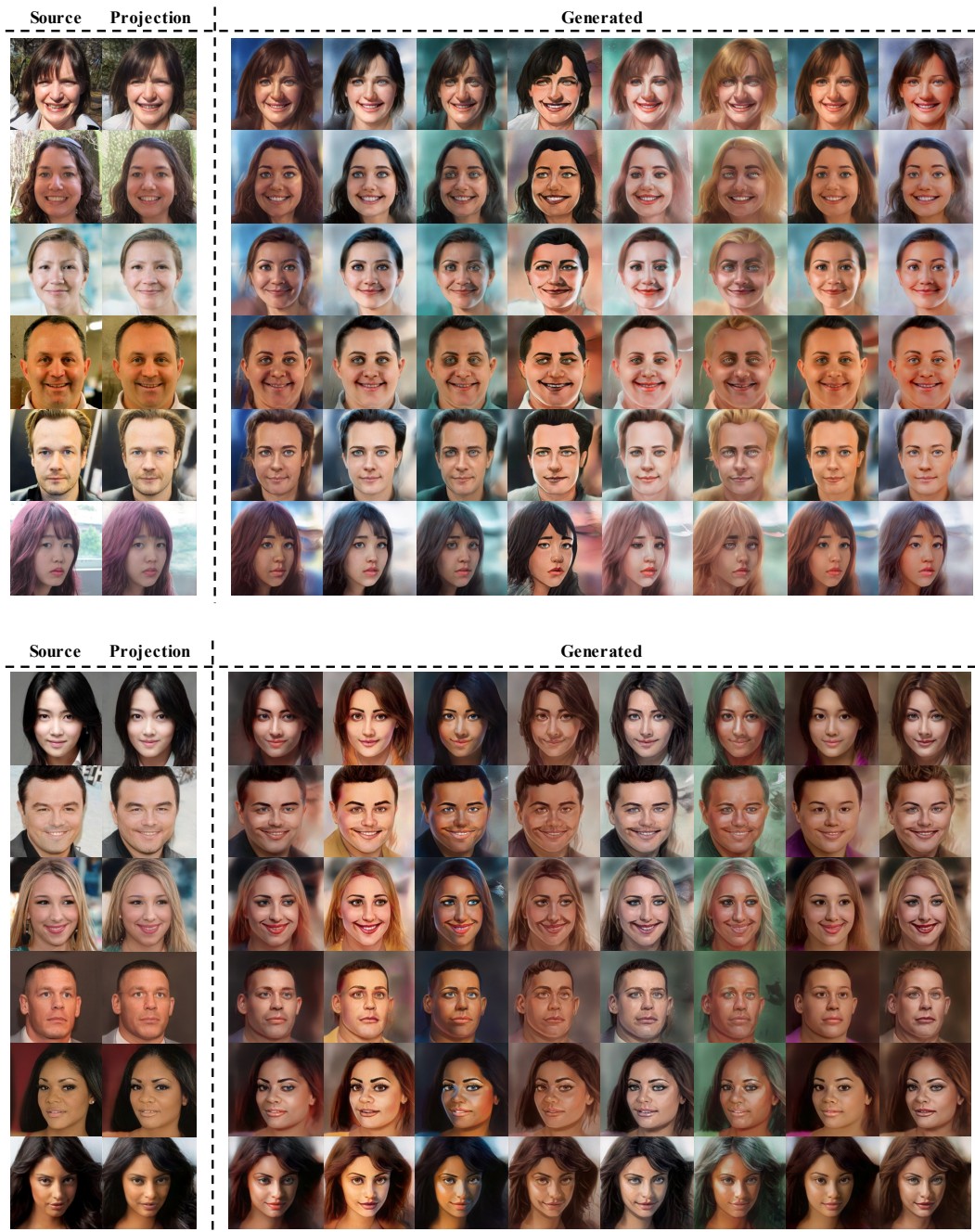

Figure 2: Latent-guided results of arbitrary style transfer. Our BlendGAN is cooperated with a pSp encoder to extract the latent codes of given face images. The source images in the upper subfigure are sampled from the FFHQ dataset, and those in the lower subfigure are sampled from the CelebA-HQ [8] dataset. The style latents are randomly sampled from $\mathcal{N}(0, 1)$, and the stylized face images are generated with indicator $i = 6$.

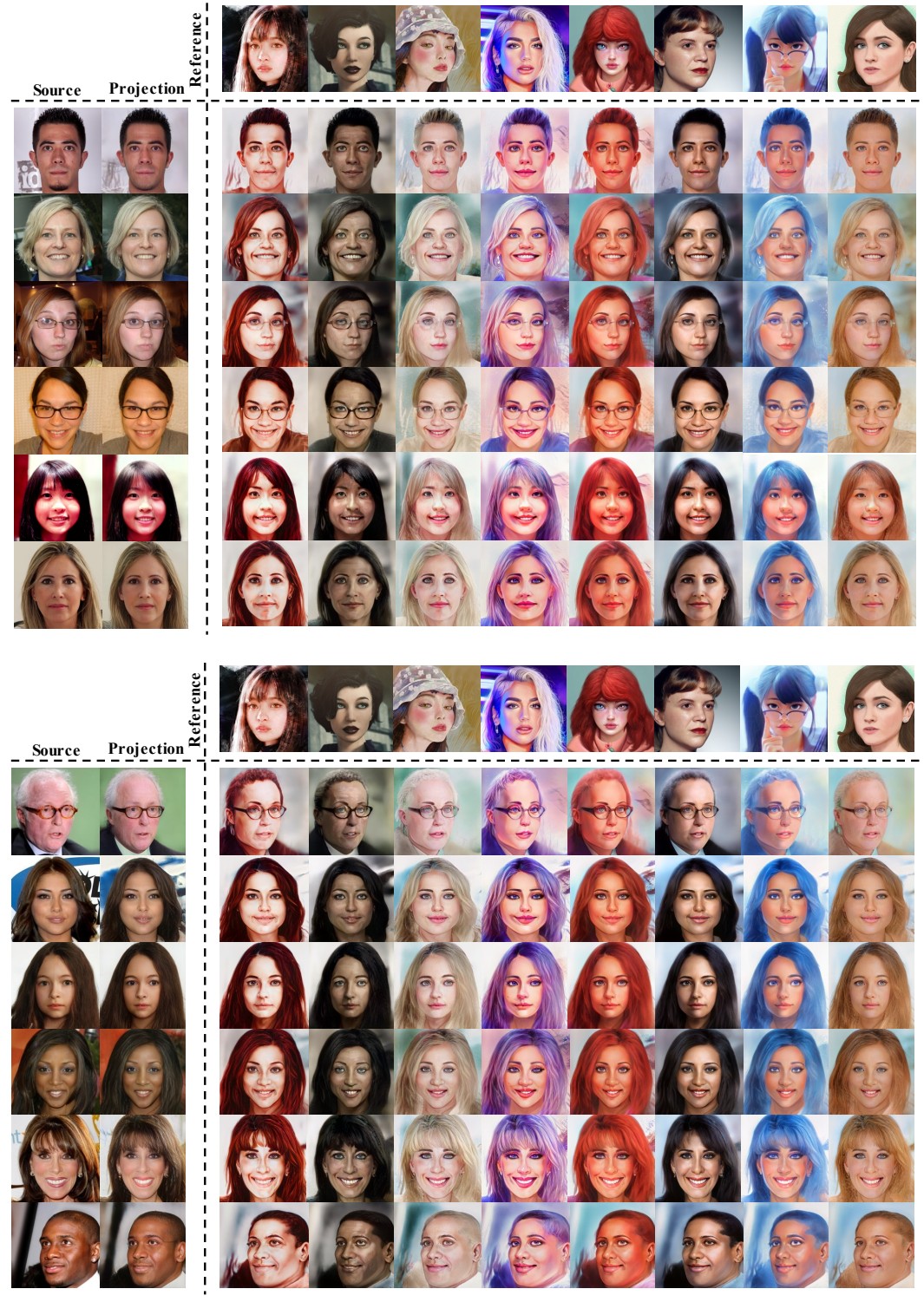

Figure 3: Reference-guided results of arbitrary style transfer. The source images in the upper subfigure are sampled from the FFHQ dataset, and those in the lower subfigure are sampled from the CelebA-HQ [8] dataset. The reference images are sampled from the AAHQ dataset, and the stylized face images are generated with indicator $i = 6$.

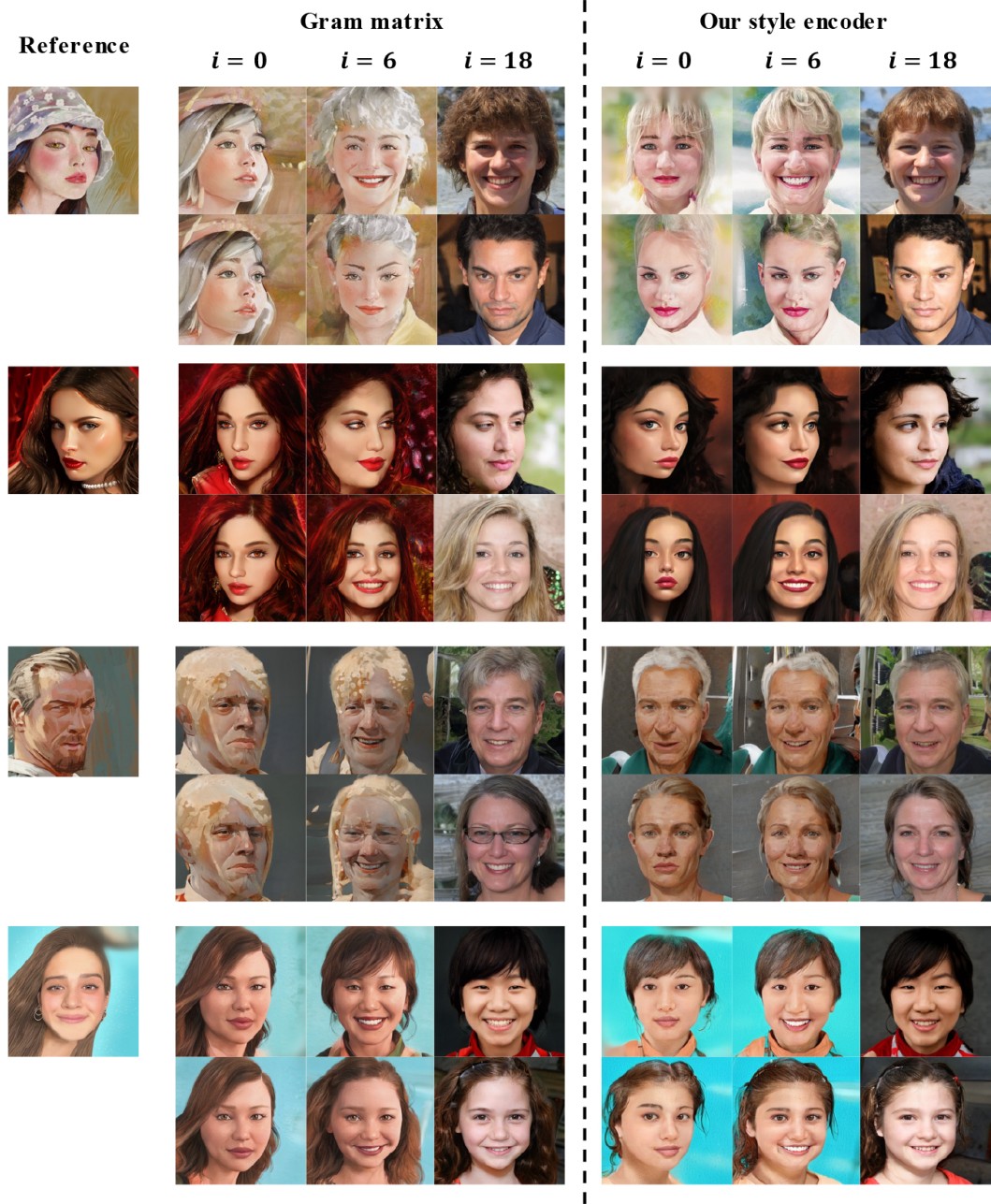

Figure 4: Comparison of two style embedding methods. Left: directly concatenating the Gram matrices without dimension reduction. Right: our self-supervised style encoder. Each row shows the generated results of a certain face latent code.

# E  Face Generation for Out-of-Distribution Style

Since our BlendGAN is trained on the AAHQ dataset, which covers a wide variety of image styles, the model has a good ability for out-of-distribution generalization. Specifically, as shown in the main text, for a reference style which is in the distribution of AAHQ, our model can easily generate corresponding stylized face images. However, for a reference image whose style is significantly different from that in AAHQ, if directly feeding it into BlendGAN, the style of generated images may not be similar to the reference. Nevertheless, our BlendGAN can perform as a style prior, and we only need to finetune the model with the reference image for a few iterations, then the resulting model can generate images with reference style. During finetuning, weights of the style encoder are fixed while those of the generator and the three discriminators are optimized. $D_{face}$ still receives images sampled from FFHQ, while $D_{style}$ and $D_{style\_latent}$ only receive the reference image. We finetune the framework for 1000 iterations with a batch size of 1, which costs about 15 minutes on a Tesla V100 GPU. The above process can be regarded as one-shot learning.

Figure 5 shows the generation results of four out-of-distribution styles, in which the reference images are illustrated by Littlethunder[4], Morechun[5], MarinaRen[6] and Dahye[7]. These four reference styles are far different from those in the AAHQ dataset, hence the resulting styles generated by the original BlendGAN are inconsistent with the reference styles. In contrast, after one-shot finetuning, the generated images have similar styles to their reference images, demonstrating the ability of our framework for out-of-distribution generalization.

# F  Additional Qualitative Results

Figure 6 and 7 illustrate additional results of our BlendGAN for latent-guided and reference-guided synthesis respectively.

---

[4]https://www.instagram.com/p/BSvQRCqFyYu

[5]http://morechun.com

[6]https://www.zcool.com.cn/work/ZNDczNTU4NjQ=.html

[7]https://www.instagram.com/p/B2Yj9kbHPw4

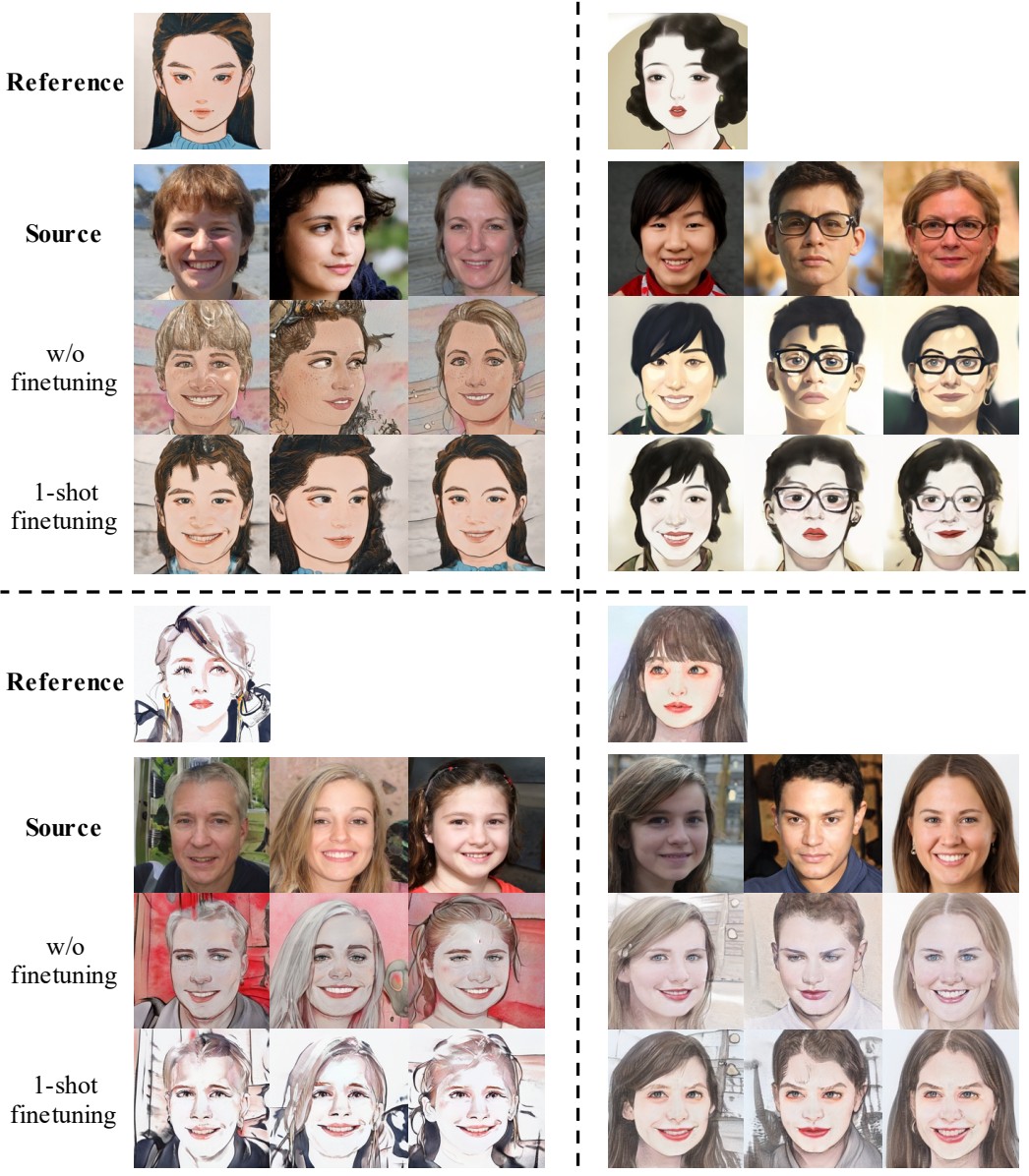

Figure 5: Comparison of face generation for out-of-distribution styles. Results of the one-shot finetuning strategy perform better style consistency than original BlendGAN.

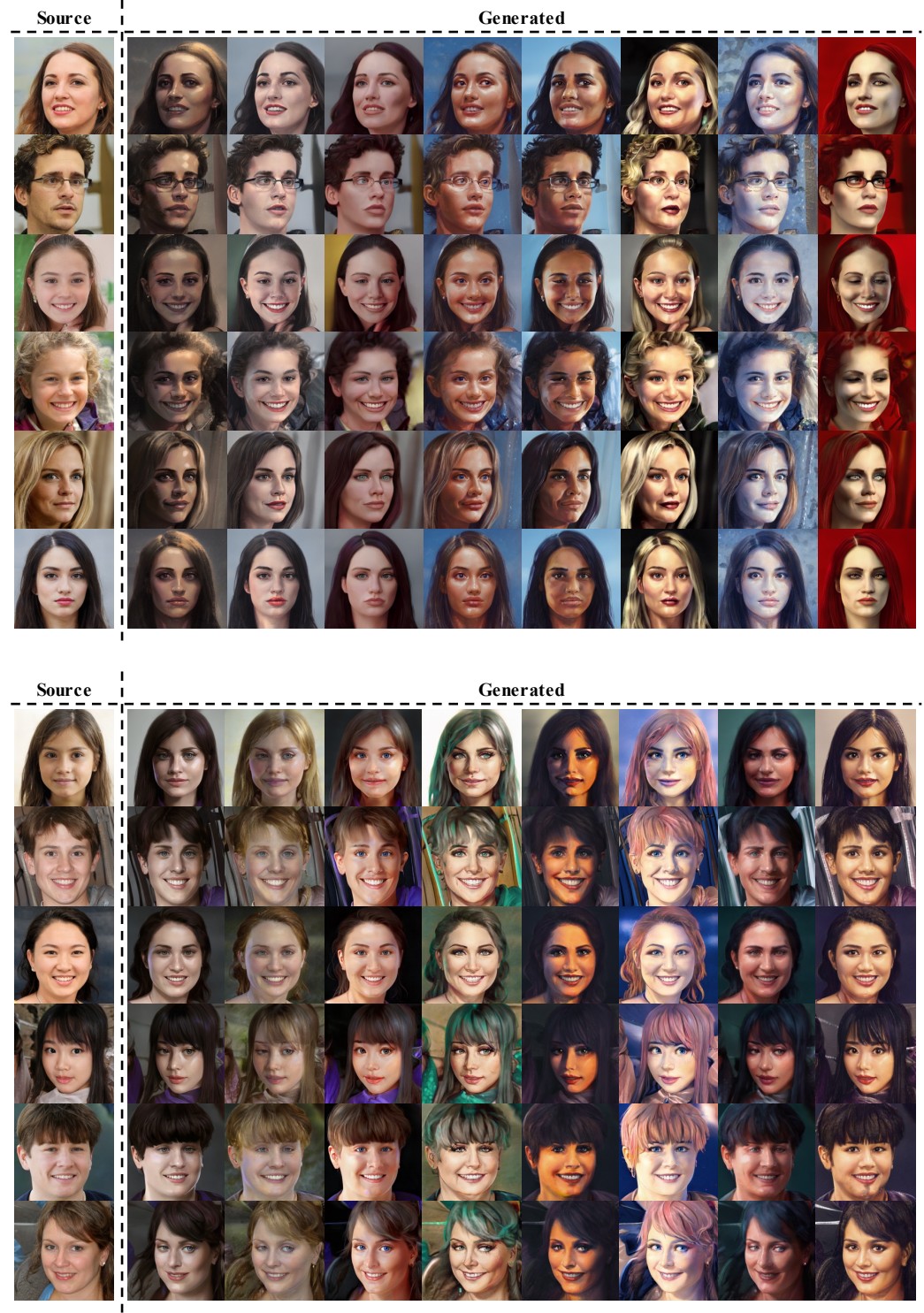

Figure 6: Additional results for latent-guided synthesis. The source images are generated by our model with indicator $i = 18$, and the style latents are randomly sampled from $\mathcal{N}(0, 1)$.

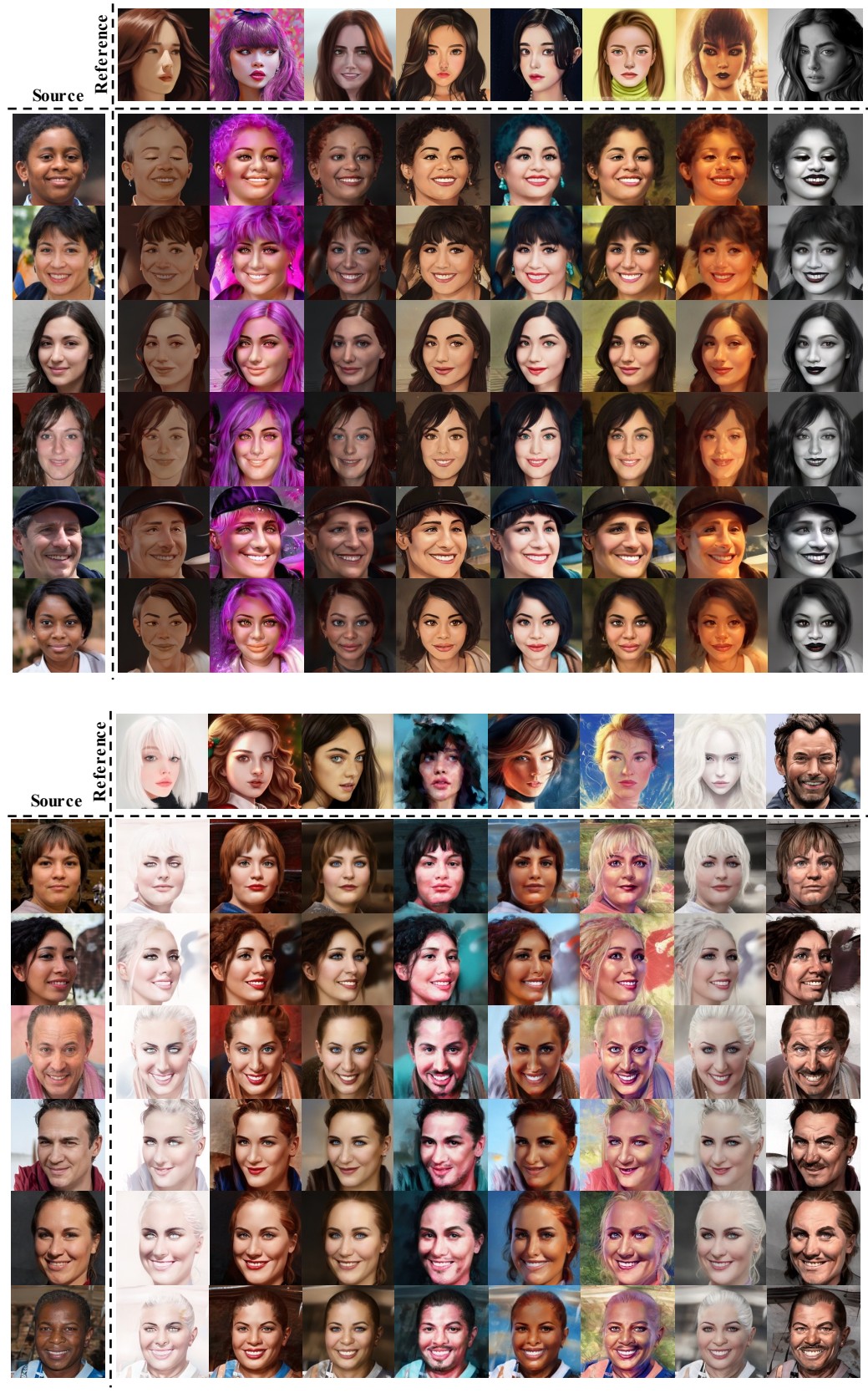

Figure 7: Additional results for reference-guided synthesis. The source images are generated by our model with indicator $i = 18$, and the reference images are sampled from the AAHQ dataset.