# OpenReview forum: "BlendGAN: Implicitly GAN Blending for Arbitrary Stylized Face Generation"
_NeurIPS.cc/2021/Conference — NeurIPS 2021 Poster_

### Official Review · Reviewer_19Vx · 2021-07-06

**Rating:** 6
**Confidence:** 4

**Summary:**

The authors propose BlendGAN to perform arbitrary multi modal face stylization. They do so by pretraining a Style Encoder on an artistic dataset, then using weighted blending module (WBM) to blend face and style representations into an image using a StyleGAN-based decoder. They outperform previous methods qualitatively and quantitatively.

**Limitations And Societal Impact:**

yes

**Main Review:**

One advantage of BlendGAN is its flexibility in using either randomly sampled style or a reference style to produce multimodal images. This allows arbitrary generation and unlike Toonify [1], does not require a new model for each new style.

I think the main novelty comes from learning two different distributions within a single StyleGAN. The style and face latent codes have its own separate distribution and when fed into the generator, with appropriate I, the generator can produce either real or artistic face. The use of WBM simply interpolates these 2 codes to generate a hybrid face.

In L143, it says i=0 corresponds to stylized face but in L151 natural faces are paired with i=0? Should this not be i=18?

Although Toonify is not multimodal, this is the closest baseline and I think it should be compared to. Otherwise the comparisons and quantitative results are sufficient and significant.

Qualitatively, BlendGAN captures textures well, but not global structures. For example, in Figure 1, all the face shapes are the same. This is a limitation due to the usage of layer swapping to main semantics. In Figure 5 if layer swapping is not used, it is closer to the style of the reference image. However, the semantics (such as expressions) are slightly off. Layer swapping is used to capture the semantics, at the cost of style similarity. The trade off seen in BlendGAN is the same as in Toonify. In this regard, there are no improvements over it.

Overall, idea is not novel and is a combination of several previous work combining AdaIN [2], Toonify [1], and SimCLR [3]. However, there are some novel concepts in the implementation of the generator. The results are overall better than previous approaches, but BlendGAN still suffers similar problems faced by those approaches.

[1] Pinkney, Justin NM, and Doron Adler. "Resolution Dependent GAN Interpolation for Controllable Image Synthesis Between Domains." arXiv preprint arXiv:2010.05334 (2020).

[2] Huang, Xun, and Serge Belongie. "Arbitrary style transfer in real-time with adaptive instance normalization." In Proceedings of the IEEE International Conference on Computer Vision, pp. 1501-1510. 2017.

[3] Chen, Ting, Simon Kornblith, Mohammad Norouzi, and Geoffrey Hinton. "A simple framework for contrastive learning of visual representations." In International conference on machine learning, pp. 1597-1607. PMLR, 2020.


**Time Spent Reviewing:**

1.5

---

> ### Author Response · Authors · 2021-08-10
> **Response to Reviewer 19Vx**
>
> We sincerely appreciate your careful and thoughtful comments and time. We will explain your concerns point by point.
>
> #### **Q1: The mistake in L151.**
>
> **A1:** We thank the reviewer for catching this mistake, and we will modify it in the final version.
>
> #### **Q2: Although Toonify is not multimodal, this is the closest baseline and I think it should be compared to.**
>
> **A2:**  As described in the third paragraph of Section 1, for a specific style, although Toonify is able to generate stylized faces, it still requires hundreds of target-style images for finetuning the StyleGAN model to avoid mode collapse. In contrast, after trained on the AAHQ dataset, our method only requires one artistic image to generate images with a specific style. In the experiments, each reference style only includes one artistic image, so it is impossible to train a Toonify model for comparison. Therefore, we did not include Toonify in the experiments section.
>
> #### **Q3: BlendGAN captures textures well, but not global structures.**
>
> **A3:** The goal of our BlendGAN is to generate arbitrary stylized faces as well as their unstylized counterparts. In this regard, changing the global structures of the stylized images will break the identity consistency between the generated natural and stylized face image pairs.
>
>
> #### **Q4: Improvement over Toonify.**
>
> **A4:** From the model aspect, our "blending" operation is significantly different from "layer swapping" in Toonify. In particular, "layer swapping" directly takes different resolution layers from the different models and combines them. In contrast, our "blending" operation is designed as a weighted blending module that implicitly combines the face and style latent codes rather than layers.
>
> From the task aspect, as described in the third paragraph of Section 1, Toonify has to train models in a case-by-case flavor, thus a single model can only generate images with a specific style and it requires hundreds of target-style images for finetuning. In contrast, our BlendGAN can gracefully fit arbitrary styles in a unified model while avoiding case-by-case preparation of style-consistent training images. Besides, our BlendGAN can also generate in a latent-guided manner, which does not even require any reference images but only randomly sampled style latent codes to generate face images with new styles.
>
> #### **Q5: About novelty.**
>
> **A5:** Thanks for bringing up the concerns. We believe “a new use of an existing technique with different goals” is meaningful when the new use is effective and brings a new perspective.
>
> 1. AdaIN
>
>    The arbitrary style transfer method AdaIN [1] introduces an adaptive instance normalization layer to modulate the deep features with the style representation. Although this method could transfer images to arbitrary styles, it only transfers the global styles of the references without considering local semantic styles, which may sometimes exhibit texture artifacts in the outputs. Unlike AdaIN, our method proposes a self-supervised style encoder as well as an image generation framework to generate arbitrary stylized faces. With the help of the well-designed architecture, our method can not only transfer the global style, but also preserve the semantic style information to some extent. As shown in Figure 7, the results of our method have much higher quality and fewer artifacts than those of AdaIN.
>
> 2. Toonify
>
>    As discussed in **A4**, there are two differences between Toonify [2] and our method.
>
>    Firstly, "layer swapping" directly takes different resolution layers from the different models and combines them. In contrast, our "blending" operation implicitly combines the face and style in the space of latent codes rather than layers.
>
>    Secondly, Toonify needs to train models in a case-by-case flavor and it requires hundreds of target-style images for finetuning, while our method can fit arbitrary styles in a unified model and only one reference image is required for a specific style.
>
> 3. SimCLR
>
>    Although we utilize the training strategy of SimCLR [3] for our self-supervised style encoder, there are several differences between the original SimCLR and our method.
>
>    Firstly, the original SimCLR aims to extract image representations for visual recognition tasks, while our style encoder is trained to extract the style representations of reference style images.
>
>    Secondly, the augmentations in the original SimCLR include affine transformations and color transformations, however, considering that the image style is strongly related to color, we only use affine transformations in the augmentation step of our method.
>
>    Thirdly, for extracting better style representations, our method applies the self-supervised learning strategy to learn the style latent codes upon the Gram matrices from a pretrained VGG network, rather than directly trains the model from images as SimCLR.
>
>
>
> [1] Huang, Xun, and Serge Belongie. "Arbitrary style transfer in real-time with adaptive instance normalization." In Proceedings of the IEEE International Conference on Computer Vision, pp. 1501-1510. 2017.
>
> [2] Pinkney, Justin NM, and Doron Adler. "Resolution Dependent GAN Interpolation for Controllable Image Synthesis Between Domains." arXiv preprint arXiv:2010.05334 (2020).
>
> [3] Chen, Ting, Simon Kornblith, Mohammad Norouzi, and Geoffrey Hinton. "A simple framework for contrastive learning of visual representations." In International conference on machine learning, pp. 1597-1607. PMLR, 2020.

---

### Official Review · Reviewer_Jfp6 · 2021-07-15

**Rating:** 7
**Confidence:** 4

**Summary:**

The paper proposed a method for face generation across a wide range of artistic styles. The key contributions are:

1) a style encoder that maps the gram matrix of the style image to a latent space that can be consumed by the StyleGAN decoder.

2) a blending module that accomplishes style transfer with linear interpolation between the style vector and the face vector, and a leant control vector that decides the “blending factor” for different layers.

The paper also introduced a new dataset for artistic face images and demonstrated the method’s effectiveness in learning those artistic styles by a single network.

**Limitations And Societal Impact:**

In general, the quality of the results is encouraging, especially considering the method is trained to work with many different styles and can be quickly adapted to new styles that haven't been trained with. However, it is also clear that the identity in the style reference image can leak into the results. This is especially obvious in Figure 7 (supplementary). It seems the gender of the source images can often be overwritten by the reference style images (rows 3 and 4).

I wonder why the i=18 results in Figure 4 (supplementary) are different between the Gram matrix approach and the style encoder approach? Did they use different face latent codes, or is there some differences in these two approaches (the values in the leaned alpha maybe?). Clarifying this would help the reader judge the likeness of the face identity in the results.

It would be helpful if the paper elaborates on the learned 18-dim alpha vector. What are the learned values in this vector? Is there any statistical bias that favors larger/smaller values in the certain layers for style transfer? Also, some comparisons between using the learned alpha and the results of using fixed alpha (such as 0.5 for midpoint interpolation) would be helpful.

Last but not least, it is not clear to me why the method restricts itself to reference style images that contain only portraits. The proposed style encoder first computes the Gram matrix, which has been shown to work well across arbitrary subjects and will drop most of the spatial content information of those subjects. So to me, it seems non-portrait reference style images probably will also work fine. On the other hand, if the objective is to make the reference style images and the face images better aligned in the latent space for blending, then a PSPNet fashion style encoder will make more sense. I would like to hear comments from the authors.

Minor: Line 142-143 says that the generator outputs stylized images when i=0 and natural images when i=18, but the rest of the paper (line 150-153 and eq. 6) seems to suggest the opposite.

**Main Review:**

The work is novel because it studies how to use the idea of blending the feature maps of two networks for artistic face generation with “arbitrary styles”. As a downstream application of StyleGAN for faces, the generated stylized images are in general of higher quality in comparison to the early style transfer methods that work on general subjects. In addition, unlike the recent StyleGAN based methods such as toonify networks, which have to be trained for each new style, the proposed method trained a single network that can be applied to a wide variety of styles. The limitation of out-of-distribution style transfer is studied and addressed by one-shot learning that fine-tunes the networks on a single out-of-distribution reference style image for 1000 iterations.

The training objectives seem intuitive, and there are sufficient technical details for reproduction. The proposed AAHQ dataset will be very interesting to the research community. I hope the authors can open it to the public.

The results are encouraging, though I have a few comments in the following section.

Overall the paper proposed a novel idea, and the method and results are solid.

**Time Spent Reviewing:**

3

---

> ### Author Response · Authors · 2021-08-10
> **Response to Reviewer Jfp6**
>
> We sincerely appreciate your careful and thoughtful comments and time. We will explain your concerns point by point.
>
> #### **Q1: The identity in the style reference image can leak into the results.**
>
> **A1:** As responded in **A1** to **Reviewer xcb2**, we do not use a "reconstruction loss" to train models to extract the style and identity codes of images simultaneously, but independently extract the style representation of artistic images by calculating the Gram matrices and utilizing a self-supervised learning strategy. Although it is difficult to completely remove the identity information of the style image while only retaining the style representation, our method still performs better than MUNIT, DRIT++ and StarGANv2 as illustrated in Figure 7 of the main paper.
>
> #### **Q2: Why are the $i=18$ results in Figure 4 (supplementary) are different between the Gram matrix approach and the style encoder approach?**
>
> **A2:**  We thank the reviewer for pointing out this discrepancy. Models of the two style embedding methods are finetuned from a pretrained StyleGAN model separately and all the parameters of the generator are updated. Due to the independent learning process during finetuning, the parameters in the two models gradually become different. Therefore, after the finetunings, even if the same latent code is fed into the two models, the generated images cannot be exactly the same, which makes the $i=18$​​​ results in Figure 4 (supplementary) different between the two approaches.
>
> #### **Q3: It would be helpful if the paper elaborates on the learned 18-dim alpha vector.**
>
> **A3:** We have listed the learned values of $\alpha$ in the table below. As the table shows, most of the first six weights (corresponding to layers 0-5) are smaller than the rest (corresponding to layers 6-17), indicating that low-resolution layers are more responsible for the face identity while high-resolution layers play a more important role for the style.
>
>
> |  Index  | 0 | 1 | 2    | 3 | 4    | 5 | 6    | 7 | 8    |
> | ---- | ---- | ---- | ---- | ---- | ---- | ---- | ---- | ---- | ---- |
> | Value |0.6138|0.0093|0.0277|0.1278|0.7315|0.3661|0.9461|0.4115|0.6414|
>
> |  Index  | 9 | 10 | 11 | 12   | 13 | 14   | 15 | 16   | 17 |
> | ---- | ---- | ---- | ---- | ---- | ---- | ---- | ---- | ---- | ---- |
> | Value |0.4607|0.9659|0.4852|0.9735|0.7110|0.8527|0.6249|0.8576|0.7189|
>
> #### **Q4: Why does the method restrict itself to reference style images that contain only portraits.**
>
> **A4:** Though non-portrait style images work well for neural style transfer approaches, it is problematic to use them under our framework due to the following two reasons.
>
> 1. Non-portrait style images are suitable for inducing global abstract style but lack the ability to provide semantic style information as portrait style images do. For example, in the last row of Figure 7, the portrait style image has black hair, red lips, and smooth white skin, our method well transfers the above semantic style information and generates a stylized face with the same characteristics.
>
> 2. If the non-portrait style images are used in our framework, the style discriminator may easily tell between a synthesised stylized face image and a non-portrait style image simply based on the content rather than the style.
>
> If we understand correctly, the *PSPNet* you mentioned refers to the pSp framework [1] rather than the semantic segmentation framework PSPNet [2]. The pSp framework is designed for reconstruction tasks to extract the face latent code, it is inappropriate for extracting the style representation. In contrast, our style encoder achieves style extraction by utilizing Gram matrices as coarse style representation and refining it to a 512-D vector with a self-supervised learning strategy.
>
> #### **Q5: Minor comments.**
>
> **A5:** We thank the reviewer for catching these mistakes, and we will modify them in the final version.
>
>
>
> [1] Elad Richardson, Yuval Alaluf, Or Patashnik, Yotam Nitzan, Yaniv Azar, Stav Shapiro, and Daniel Cohen-Or. Encoding in style: a stylegan encoder for image-to-image translation. *arXiv preprint arXiv:2008.00951*, 2020.
>
> [2] Zhao, H., Shi, J., Qi, X., Wang, X., & Jia, J. (2017). Pyramid scene parsing network. In *Proceedings of the IEEE conference on computer vision and pattern recognition* (pp. 2881-2890).

---

> > ### Comment · Reviewer_Jfp6 · 2021-08-20
> > **Response to author's feedback**
> >
> > I read the feedback and all of my previous comments have been addressed. Thanks for the authors' effort.
> >
> > By taking into consideration of both authors' feedback and reviewers' comments, I keep my original score (7)

---

### Official Review · Reviewer_hywe · 2021-07-16

**Rating:** 6
**Confidence:** 4

**Summary:**

This work proposes a method to synthesize faces with arbitrary artistic styles, where the style can be obtained from random samples or a reference style of a real face image. To achieve this, the authors introduce a style encoder to predict the style latent of a reference image. A face latent which controls the structure of the face is randomly sampled. The style latent and face latent are mixed in a fashion of style-mixing to generate the final output, where the degree of stylization is controlled by the number of layers blended with the style latent. The model is trained on FFHQ and a stylized face dataset the authors introduced simultaneously, where the face latent correspond to

**Limitations And Societal Impact:**

1. Prior work by Park et.al. [1] also provides a solution to the proposed problem but is not compared or discussed in this work. Park et.al. show capabilities of arbitrary stylization in multiple domains (e.g. LSUN church, FFHQ, AFHQ, etc.), while the proposed method conducted experiments on only the face domain. Also, Park et.al. allows structure and style separation by training on a single domain, while the proposed method requires collecting a real and stylized domain separately. It appears that Park et.al. provides a stronger yet more elegant solution to this problem.

2. No actual real image editing is shown in this work. Although real stylized images are used as a reference for the style, it is also important to show the capability of taking a real source image and stylize it.

3. In evaluation, all the source images are face images generated by the proposed method without stylization. These source images are then used as the inputs for the baselines. Since the baselines are image-to-image translation methods, the comparison is fairer if actual real images are used as the source images.

4. I'm confused why the completely stylized image ($i=0$) shares the same structure as the other faces ($i>0$). Does any of the face latent $w_f$ got blended into the final output? My guess is that the $\theta$ term in Eq. 5 allows this preservation, but will be great to verify this.

[1] Park et.al. Swapping Autoencoder for Deep Image Manipulation.

**Main Review:**

This work provides a clear explanation of their method, and the writing is easy to follow. Also, the proposed dataset (AAHQ) will be useful to the community. Overall, the proposed idea of allowing arbitrary style input and style blending is interesting. However, both ideas have been explored by prior work. Also, while most prior work uses real images as sources, the proposed method only shows results on synthesized source images. It will be more convincing to provide real image editing results, so this work can be applied in a more realistic scenario. In my opinion, these issues will need to be addressed to make this work strong. More details are provided in the limitation section.

Update: I read the feedback and all of my previous comments have been addressed. Also, after reading other reviewers' comments and authors' feedback, I agree that this work provides a novel way to combine stylization and generation, while also improving upon prior works. I will change my ratings to a 6.

**Time Spent Reviewing:**

3.5

---

> ### Author Response · Authors · 2021-08-10
> **Response to Reviewer hywe**
>
> We sincerely appreciate your careful and thoughtful comments and time. We will explain your concerns point by point.
>
> #### **Q1: Prior work by Park et.al. provides a stronger yet more elegant solution to this problem.**
>
> **A1:** The work by Park et.al. [1] shows good translation results for given images, and we will cite this work in the final version. However, there are several differences between this method and ours as follows:
>
> 1. The goal of our method is arbitrary stylized face generation. By utilizing a StyleGAN-based architecture, our method can generate infinite stylized faces with reference style as well as their unstylized counterparts, which can be used for a variety of downstream tasks, such as image-to-image translation. The work by Park et.al. is proposed for reference-guided image-to-image translation task, and it is more similar to DRIT++ [2] and the *reference-guided synthesis* part of StarGANv2 [3]. As discussed in the second paragraph of Section 2, this type of task requires a source image and a reference image to synthesise the result, hence, the number of generated images is constrained by the number of source and reference images.
> 2. In order to generate a stylized image, Park et.al. requires an existed reference image to extract the texture code. However, since the style latent code can be randomly sampled from the standard Gaussian distribution, our method is able to create new styles without reference artistic images (as shown in Section 4.2). Therefore, images generated by our method have larger style diversity.
> 4. As the caption of Figure 7 in Park et.al. says, for continuous transformation, it needs to average the vector difference between the texture codes of two domains based on annotations from the training sets. In contrast, our method only needs to vary the value of $i$ in the WBM to perform the continuous transformation, which does not require any annotations or average calculation.
>
> #### **Q2: No actual real image editing is shown in this work.**
>
> **A2:** We have shown the stylization results of real source images in Section C and Figure 2-3 of the supplementary materials. Besides, it is also footnoted on the second page of the main paper.
>
> #### **Q3: Since the baselines are image-to-image translation methods, the comparison is fairer if actual real images are used as the source images.**
>
> **A3:** As discussed in the first term of **A1**, our method is focused on arbitrary stylized face generation. To the best of our knowledge, there are few methods that could achieve the goal consistent with ours - to generate infinite arbitrary stylized faces as well as their unstylized counterparts. The baselines are the closest methods to our goal of arbitrary stylized face generation. In this regard, using the generated images as source images are reasonable for the comparison. Besides, we have also conducted experiments for the stylization of real source images by utilizing a pSp encoder [4]. Figure 2 and Figure 3 of the supplementary materials show the results.
>
> #### **Q4: Why does the completely stylized image ($i=0$) share the same structure as the other faces ($i>0$) ?**
>
> **A4:** We would like to clarify that the generated stylized image is controlled by the face latent code and the style latent code together. When $i=0$, we have $m=[1,1,1,\ldots,1]$, so $w_{f}$ and $w_{s}$ are blended in all layers (not just $w_{s}$ works). The face latent code (corresponding to $w_{f}$) controls the face identity of the generated image. Therefore, the completely stylized image ($i=0$) shares the same structure as the other faces ($i>0$). Please refer to Equation 3-5 for more details.
>
>
>
> [1] Park et.al. Swapping Autoencoder for Deep Image Manipulation.
>
> [2] Hsin-Ying Lee, Hung-Yu Tseng, Qi Mao, Jia-Bin Huang, Yu-Ding Lu, Maneesh Kumar Singh, and Ming-Hsuan Yang. Drit++: Diverse image-to-image translation viadisentangled representa- tions. *International Journal of Computer Vision*, pages 1–16, 2020.
>
> [3] Yunjey Choi, Youngjung Uh, Jaejun Yoo, and Jung-Woo Ha. Stargan v2: Diverse image synthesis for multiple domains. In *Proceedings of the IEEE Conference on Computer Vision and Pattern Recognition*, 2020.
>
> [4] Elad Richardson, Yuval Alaluf, Or Patashnik, Yotam Nitzan, Yaniv Azar, Stav Shapiro, and Daniel Cohen-Or. Encoding in style: a stylegan encoder for image-to-image translation. *arXiv preprint arXiv:2008.00951*, 2020.

---

> > ### Comment · Reviewer_hywe · 2021-08-26
> > **Reviewer Response**
> >
> > Thanks to the author for the clarification.
> >
> > I read the feedback and all of my previous comments have been addressed. Also, after reading other reviewers' comments and authors' feedback, I agree that this work provides a novel way to combine stylization and generation, while also improving upon prior works. I will change my ratings to a 6.

---

> > > ### Author Response · Authors · 2021-08-30
> > > **Thanks for your positive feedback**
> > >
> > > Dear Reviewer hywe,
> > >
> > > We sincerely thank you for your positive feedback and will appreciate it if you can update the rating score in the system accordingly.
> > >
> > > Thanks,
> > >
> > > Authors

---

### Official Review · Reviewer_xcb2 · 2021-07-17

**Rating:** 7
**Confidence:** 3

**Summary:**

The paper introduces a styleGAN based stylization approach that allows the transfer of arbitrary style using a single model. The contribution includes: (1) the large-scale AAHQ dataset, (2) design of the style encoder using self-supervised representation learning, (3) the general idea of separating the latent code into a face code and a style code and using a weighted blending module to control how to blend the codes together to control the amount of style that get transferred into the result.

**Ethics Review Area:**

["I don’t know"]

**Main Review:**

I think the paper contains novel ideas and demonstrates compelling results. Though not the first paper to support the transfer of arbitrary style, compared to the previous feed-forward or optimization-based stylization approaches, the result quality has visibly improved. Therefore, I am in favor of accepting this paper but I do have the following suggestions and questions.

The results at a first glimpse is great. But with close examination, there are following issues.
In the supplementary material, some results indicate that identity is sometimes not well-maintained. For example, in Figure 5 bottom right corner some results show visible gender change. In figure 7, the rightmost column, women are getting beard from the style reference. These reflect that style and identity is not completely disentangled during the training. Some of the identity aspect is carried over from the style reference. Figure 6 shows that, the stylization results sometimes have different gaze of facial expression from the input (for example, the middle two results in the rightmost column have the eyes closed). Figure 7 shows that style is sometimes not preserved very well. For example, the lip color in the results is different from the style reference in the rightmost two columns and the second column from the left. I think this reflects the fundamental limitation of this type of stylization approach which is that it lacks semantic correspondence between the style reference and the result. I think the main paper should discuss these limitations.

It will be great to see ablation studies to analyze the effect of the style latent discriminator. What is the architecture of the style latent discriminator?  Several of the decision choices in the paper lack motivation and analysis. For example, the exact design of the two MLPs seems quite arbitrary. How do you decide how many layer to use?

In Line 224, the paper says “StarGANv2 … have some subtle artifacts and are not as natural as our results”. can you be more specific and point out the artifacts in StarGAN2 results? Also since the styles of StarGAN2 and the current paper are different, it is a bit hard to directly compare the results.

I think the WBM paragraph in Section 3.2 can use some higher level description and motivation before diving into the equations.

The supp mentions 128M images are used to train the style encoder. How did you obtain the images? They are augmented from the AAHQ dataset or is it a new dataset? Why do you need so many images ? Have you tried encoder training with less?  Later on, the supp mentions 1.6M images used to train the generator and discriminators. How did you obtain these images through augmenting FFHQ + AAHQ?

Figure 2 and Figure 3 in the supp mention that the source images are sampled from FFHQ dataset and the reference image from AAHQ. Are these images held off from training or are they part of the training set?




**Time Spent Reviewing:**

3

---

> ### Author Response · Authors · 2021-08-10
> **Response to Reviewer xcb2**
>
> We sincerely appreciate your careful and thoughtful comments and time. We will explain your concerns point by point.
>
> #### **Q1: Style and identity are not completely disentangled during the training.**
>
> **A1:** How to disentangle the style and identity of an image completely is a long-standing problem in the research community. Many methods have been proposed to solve this problem, such as MUNIT [1], DRIT++ [2], and StarGANv2 [3]. All of these methods try to train a model to extract the style and identity codes of images at the same time through the constraint of "reconstruction loss". This constraint requires that the original image can be reconstructed by combining the extracted style and identity codes. Our method provides a new way to preserve the face identity of the source image and extract the style of the reference image simultaneously.
>
> **Identity** In order to make the generated stylized face have better face identity consistency with the corresponding natural face, we introduce the WBM and use the indicator $i$ to control the stylization strength and the face identity consistency. After analysis, we believe that the issue you mentioned that the identity is sometimes not well-maintained is caused by the fact that we use $i=6$ for all the reference styles. Though a fixed setting is not always the perfect choice for different styles, in Section 4.1 we show through experiments that $i=6$ is a good tradeoff between the stylization strength and the face identity consistency.
>
> **Style** In this work, we do not use a "reconstruction loss" to train models to extract the style and identity codes of images, but independently extract the style representation of artistic images by calculating the Gram matrices and utilizing a self-supervised learning strategy. Although it is difficult to completely remove the identity information of the style image while only retaining the style representation, our method still performs better than MUNIT, DRIT++ and StarGANv2 as illustrated in Figure 7 of the main paper.
>
> #### **Q2: This type of stylization approach lacks semantic correspondence between the style reference and the result.**
>
> **A2:** Due to the same issue discussed in the **Style** part of **A1**, our style encoder is still a little far from perfectly retaining all the style representation with semantic correspondence. Although several regions in the result images may show an inconsistent style as the reference images, most of our result images perform well for semantic correspondence, such as the hair color and skin texture in Figure 1 of the main paper and Figure 7 of the supplementary materials.
>
> #### **Q3: It will be great to see ablation studies to analyze the effect of the style latent discriminator.**
>
> **A3:** As discussed in Section 3, the style latent discriminator $D_{style\_latent}$ is the only part in our framework that constrains the style consistency between the generated image $x_s$ and the style latent code $z_s$. Hence, if $D_{style\_latent}$ is removed, the style of the generated stylized-face image will be random and will not be consistent with the style latent code. In fact, we have conducted an experiment without the style latent discriminator, and the results also prove the above statement.
>
> #### **Q4: The exact design of the architecture.**
>
> **A4:**
>
> - The style latent discriminator is essentially a conditional discriminator which is often used in conditional GANs, and it can adopt many different architectures. In our experiments, we choose the architecture of *projection discriminator* [4], which has also been used in SNGAN [5], SAGAN [6], and BigGAN [7].
> - The two MLPs used in our generator have the same architecture as in StyleGAN [8] and StyleGAN2 [9]. We followed the default layer number they set for the MLP. In Table 4 of the StyleGAN paper, it shows that the generator has the best performance when the MLP has 8 layers.
>
>
> #### **Q5: In Figure 6, can you be more specific and point out the artifacts in StarGANv2 results? Also since the styles of StarGANv2 and the current paper are different, it is a bit hard to directly compare the results.**
>
> **A5:**
>
> - Among the results generated by StarGANv2 in Figure 6, the upper part of the glasses frame is fused with the eyebrows in the second face, and the last face has a slight blob-shaped artifact on the right side of the forehead.
> - As for each row of the generated images in Figure 6, the style latent used by each method is different. We just use Figure 6 to show the general image quality of each method for latent-guided generation, readers could refer to Figure 7 to compare the results of each method in the same style.
>
> #### **Q6: The WBM paragraph in Section 3.2 can use some higher level description and motivation before diving into the equations.**
>
> **A6:** As mentioned in Section 3.2, the WBM is proposed based on the characteristic of StyleGAN that different resolution layers in the model are responsible for different features in the generated image, and it performs as a controller to decide in which layers the style latent codes should be blended into the face latent codes. In this way, the generator could synthesise natural and stylized face pairs with the same face identity. After the equations, we also used examples to explain how it works.
>
>
> #### **Q7: About the numbers of training images mentioned in the supplementary materials.**
>
> **A7:** We suppose that the reviewer may have misunderstood our description of the image numbers for training the style encoder and the generator. In Section A of the supplementary materials, we refer to the description manners in the StyleGAN papers \[8] [9]. The numbers 128M and 1.6M mean the models are shown a total of 128M and 1.6M real images during the whole training procedure, rather than the sizes of training datasets. For example, when training the style encoder, we use a minibatch size of 256 and train the model for 500k iterations, so 256 * 500k = **128M** images are shown to the model, this number corresponds to the **128M** described in the supp.
>
> #### **Q8: Figure 2 and Figure 3 in the supp mention that the source images are sampled from FFHQ dataset and the reference image from AAHQ. Are these images held off from training or are they part of the training set?**
>
> **A8:** We randomly sample these images from the training set to perform as an experiment of "in-domain" arbitrary style transfer. For "out-of-domain" experiments, people could refer to image results in Section E. In fact, we have also tested our method on CelebA-HQ dataset and get the same good results. We will add these results in the final version if required.
>
>
>
> [1] Xun Huang, Ming-Yu Liu, Serge Belongie, and Jan Kautz. Multimodal unsupervised image-to-image translation. In *Proceedings of the European conference on computer vision (ECCV)*, pages 172–189, 2018.
>
> [2] Hsin-Ying Lee, Hung-Yu Tseng, Qi Mao, Jia-Bin Huang, Yu-Ding Lu, Maneesh Kumar Singh, and Ming-Hsuan Yang. Drit++: Diverse image-to-image translation viadisentangled representations. International Journal of Computer Vision, pages 1–16, 2020.
>
> [3] Yunjey Choi, Youngjung Uh, Jaejun Yoo, and Jung-Woo Ha. Stargan v2: Diverse image synthesis for multiple domains. In Proceedings of the IEEE Conference on Computer Vision and Pattern Recognition, 2020.
>
> [4] Takeru Miyato and Masanori Koyama. cgans with projection discriminator. In *International Conference on Learning Representations*, 2018.
>
> [5] Takeru Miyato, Toshiki Kataoka, Masanori Koyama, and Yuichi Yoshida. Spectral normalization for generative adversarial networks. In *International Conference on Learning Representations*, 2018.
>
> [6] Han Zhang, Ian Goodfellow, Dimitris Metaxas, and Augustus Odena. Self-attention generative adversarial networks. In *International conference on machine learning*, pages 7354–7363. PMLR, 2019.
>
> [7] Andrew Brock, Jeff Donahue, and Karen Simonyan. Large scale gan training for high fidelity natural image synthesis. In *International Conference on Learning Representations*, 2018.
>
> [8] Tero Karras, Samuli Laine, and Timo Aila. A style-based generator architecture for generative adversarial networks. In *Proceedings of the IEEE/CVF Conference on Computer Vision and Pattern Recognition*, pages 4401–4410, 2019.
>
> [9] Tero Karras, Samuli Laine, Miika Aittala, Janne Hellsten, Jaakko Lehtinen, and Timo Aila. Analyzing and improving the image quality of stylegan. In *Proceedings of the IEEE/CVF Conference on Computer Vision and Pattern Recognition*, pages 8110–8119, 2020.

---

> > ### Comment · Reviewer_xcb2 · 2021-08-26
> > **Recommend acceptance**
> >
> > Thanks for the detailed answers for my questions. I remain positive about the paper and recommend acceptance.

---

### Decision · Program_Chairs · 2021-09-27

**Decision:**

Accept (Poster)

**Comment:**

UPDATE: The revision of this paper has been reviewed and the paper has been accepted.  However, the following additional minor changes are required for the camera-ready:
- Acknowledge in the main text the issue that the new model has the same challenges as PULSE (i.e., input images with darker skin are now stylized as lighter skinned faces in the output).
- Update the "Broader Impacts" section to mention the fact that face data is biometric data and thus needs to be sourced and distributed more carefully, with attention to potential privacy, consent and copyright issues.

----

It came to the attention of the program chairs and ethics review chairs very late in the review process that this paper deals with face generation, a sensitive application, but was not flagged for ethics review.  An emergency ethics review was obtained and the program chairs and ethics review chairs then discussed the paper in detail.

Based on this discussion, we have decided to conditionally accept the paper, given the ethical concerns related to face generation more broadly. In order for this paper to be fully accepted, authors need to address the following concerns:

- Provide examples beyond light-skinned faces and other potentially biased results or demonstrate some mitigation and reflection on any biased outcomes of the model. Include an analysis or disaggregated evaluation acknowledging any limitations in how the model handles faces of differing demographics.
- Communicate any face data distribution restrictions or limitations to model distribution in light of privacy or malicious use concerns.
- Acknowledge any potential harmful applications or depictions that could arise from the use of this technology in an expanded “broader impacts and ethical considerations” section.

We appreciate the cooperation of the authors in this process, and we hope that these adjustments and further reflection will improve the overall quality of the work. We hope authors can commit to these improvements as a requirement for inclusion at this conference.

The original meta-review from the AC follows.

----

This paper proposes a model for synthesizing faces with diverse styles. The model can generate arbitrary stylized faces and their natural photo versions at the same time. In general, many reviewers find the idea interesting and the results encouraging. There are concerns regarding the technical novelty and comparisons against Swapping Autoencoder and Toonify. The rebuttal addressed most of the concerns and clarified the difference between the proposed work vs. Toonify and Swapping Autoencoder. The AC agreed with the reviewers’ consensus and recommended accepting the paper.